# Detection of Potential Markers for Lip Vermilion Epithelium in Japanese Macaques Based on the Results of Gene Expression Profile

**Hiroko Kato** [1], **Yiwei Ling** [2], **Emi Hoshikawa** [1,3], **Ayako Suzuki** [1], **Kenta Haga** [1], **Eriko Naito** [1,4], **Atsushi Uenoyama** [4], **Shujiro Okuda** [2] **and Kenji Izumi** [1,*]

1   Division of Biomimetics, Faculty of Dentistry & Graduate School of Medical and Dental Sciences, Niigata University, Niigata 951-8514, Japan; kato-hi@phs.osaka-u.ac.jp (H.K.); hoshikawa@dent.niigata-u.ac.jp (E.H.); suzuki-a@dent.niigata-u.ac.jp (A.S.); haga@dent.niigata-u.ac.jp (K.H.); n-eriko@dent.niigata-u.ac.jp (E.N.)
2   Division of Bioinformatics, Graduate School of Medical and Dental Sciences, Niigata University, Niigata 951-8514, Japan; seraphwyl@med.niigata-u.ac.jp (Y.L.); okd@med.niigata-u.ac.jp (S.O.)
3   Division of Periodontology, Faculty of Dentistry & Graduate School of Medical and Dental Sciences, Niigata University, Niigata 951-8514, Japan
4   Division of Oral and Maxillofacial Surgery, Faculty of Dentistry & Graduate School of Medical and Dental Sciences, Niigata University, Niigata 951-8514, Japan; au@dent.niigata-u.ac.jp
*   Correspondence: izumik@dent.niigata-u.ac.jp; Tel.: +81-25-227-2850

**Abstract:** Development of effective in vitro human lip models, specific to the vermilion epithelium, has not progressed as much as that of skin and oral mucosa/gingiva models in vitro. Our histologic examination demonstrated that a Japanese macaque (male, 7 years and 9 months old) had vermilion in the lip distinct from adjacent skin and oral mucosa, resembling histological characteristics of the human lip. Therefore, in this study, we examined the gene expression profile of the three distinct epithelia (skin/vermilion/oral mucosa) within the lip of a Japanese macaque to explore a single potential marker of human vermilion epithelium. Six pairwise comparisons in the skin/vermilion/oral mucosa epithelium in vitro and in vivo revealed 69 differentially up-regulated genes in vermilion epithelium in vivo, in which a few unique genes were highly expressed when compared with both skin and oral mucosa epithelium in vivo using clustering analysis. However, we could not detect a single marker specific to vermilion epithelium supported by the gene expression profile of a Japanese macaque. Instead, the pair of keratin 10 and small proline-rich protein 3 resulted in a potential marker of vermilion epithelium in the human lip (female, 53-year-old) via a double-immunostaining technique. Nonetheless, our result may provide further clues leading to other potential markers of the vermilion epithelium.

**Keywords:** lip; vermilion epithelium; gene expression profile; potential marker; Japanese macaque; in vivo; keratinocytes

## 1. Introduction

The lip is a unique tissue anatomically and histologically in humans for a transition zone, which is referred to as vermilion and is located between keratinized skin and non-keratinized labial oral mucosa [1,2]. The skin is covered by a thin, stratified squamous keratinized epithelium involved in hair follicles. The skin transitions to the vermillion zone, forming the red portion of the lip, and continues to the moist oral mucosa. The stratified epithelium continues to become thicker across the lip (from vermillion to oral mucosa) [3]. There are no adnexal structures (hair follicles and sweat glands) in the vermilion dermis [1]. The numerous, densely-packed dermal papillae of the lamina propria underlying the vermillion epithelium allow blood vessels close access to the surface, conferring a red color to vermillion. Minor salivary glands are located in the underlying tissue of the oral mucosa. Orbicularis oris muscle presents at the central core of the lip [1,4]. In addition

to the distinct epithelial morphology of the human lip (skin/vermillion/oral mucosa), the histological characteristics are closely pertinent to its barrier function because the lip is located at the boundary of dry and moist conditions caused by severe environmental damages [5]. Therefore, it is useful to produce an efficient cellular model to investigate the correlation of the characteristics of the lip epithelium with the barrier function under cellular and molecular bases [6,7]. Additionally, the model can be used to evaluate a route via the vermilion that can be another drug delivery system due to those unique tissue properties of the lip [4,8].

Our final goal is to develop a specific human lip/vermilion epithelium in vitro model to assess the efficacy and safety of various pharmaceutical products and consumer drugs for topical application to the vermilion. Due to the vermilion epithelium's unique features, compared with skin and oral mucosa, currently available in vitro skin and oral mucosa models have limitations in use when accurately evaluating and confirming the effects of testing products on the lip/vermilion epithelium [9]. Thus, there is a need to explore and detect single potential markers specific to the vermilion epithelium that differentiate from adjacent skin and oral mucosa. However, procuring a fresh human lip tissue sample that includes all epithelia that continuously originate from the skin, vermilion, and oral mucosa is particularly challenging due to aesthetic issues. Lack of sufficient tissue samples for research use can be attributable for the limitations in biological study of the lip.

According to a prior histological study, possibly for the first time, this study showed that a Japanese macaque presented with vermilion epithelium in the lip distinct from the adjacent skin and oral mucosa, which is very similar to the characteristics of the human lip. This has allowed us to obtain a sufficient amount of the lip tissue to thoroughly investigate the vermilion epithelium. Therefore, in the present study, we aimed to perform microarray analysis for in vivo tissues as well as primary keratinocytes in a culture derived from three distinct epithelia in the lip of a Japanese macaque, and examined the gene expression profile that distinguished vermilion epithelium from adjacent skin and oral mucosa epithelia. Since gene clustering analysis determines differentially expressed genes specific to the vermilion epithelium, results should contribute to detecting a single potential marker specific to the human vermilion epithelium. Furthermore, herein, we have used immunohistochemistry to test specific protein expression detected by the gene clustering analysis and their expression patterns in human and monkey lip in vivo to support the gene expression profile of the Japanese macaque.

## 2. Materials and Methods

### 2.1. Preparation for Lip Tissue of a Japanese Macaque (Skin, Vermilion and Oral Mucosa)

The procurement of the entire lip tissue of a Japanese macaque (male, 7-year and 9-month old) complied with the National Institutes of Health Guidelines for the Care and Use of Laboratory Animals. The Niigata University Institutional Animal Care and Use Committee approved the study's experimental procedures (SA00008, 31 March 2017).

The total harvested lip tissue was dissected through the orbicularis oris muscle tissue using scissors, which developed an entire section of lip tissue that consisted of the skin, lip vermilion, and labial oral mucosa. The muscle tissue was trimmed off beneath the dermis/submucosal connective tissue. The tissue was cut into three different sections of the lip, such as skin, vermilion, and oral mucosa using a scalpel. The skin was macroscopically cut off from the vermilion by differentiating the tissues that had hair from those without hair. To differentiate the vermilion from the oral mucosa, 0.1% solution of Sudan black B (Merck, Darmstadt, Germany), dissolved in 70% ethanol (Wako chemical, Osaka, Japan) followed by filtration with a 0.2 μm membrane filter (Merck, Darmstadt, Germany), was applied to the rest of the (hairless) tissue surface using a cotton swab on which the vermilion was stained in black. This resulted in three different lip tissue sections. Each tissue was dissected into several pieces, which were approximately 5 mm$^2$ in size, for the below-mentioned examinations.

## 2.2. RNA Sampling from Lip Tissue of a Japanese Macaque (Skin, Vermilion, and Oral Mucosa)

After removing the adipose tissue to the maximum extent, 1500 PU/mL of dispase II (#383-02281, Godo-shusei, Tokyo, Japan) that was dissolved in Hanks' balanced salt solution (Wako Chemical, Osaka, Japan) was intra-cutaneously injected using a 1 mL syringe with a 30-gauge needle to develop blisters over the entire surface of the piece of tissue piece. This helped detach the epithelial layer from the underlying connective tissue. The pieces of tissue, which were approximately 5–10 mm$^2$, were incubated with dispase II at 37 °C for 60 m (oral mucosa) or 30 m (skin and vermilion). Finally, the pieces of tissue were washed with cold defined phosphate buffered saline (D-PBS; Wako Chemical, Osaka, Japan), and forceps were used to separate the epithelial layer from the underlying tissue. The detached epithelial layer was washed with cold D-PBS and weighed.

For microarray analysis, the epithelial layer was homogenized in an appropriate volume of QIAzol (Qiagen, Valencia, CA, USA), 100 mg of the tissue per 1 mL in a 1.5 mL tube (Eppendorf, Tokyo, Japan) using a homogenizer (Thermo Fisher Scientific, Waltham, MA, USA), and the total RNA was extracted using an RNeasy mini kit (Qiagen, Valencia, CA, USA), according to the manufacturer's instruction.

## 2.3. Culturing Primary Lip Keratinocytes (Skin, Vermilion, and Labial Oral Mucosa)

Primary keratinocyte cultures from three sections of lip tissue were established using the explant culture technique. Small explants (dermal side up) were placed in a 60 mm Petri dish (Corning, Corning, NY, USA). They were incubated in a moist atmosphere at 37 °C and 5% $CO_2$ with complete EpiLife that contained EpiLife-defined growth supplements (Thermo Fisher Scientific, Waltham, MA, USA), 0.06 mM $Ca^{2+}$, gentamicin (5.0 µg/mL; Thermo Fisher Scientific, Waltham, MA, USA), and amphotericin B (0.375 µg/mL; Thermo Fisher Scientific, Waltham, MA, USA), which is a serum-free culture medium. The culture medium was refreshed with complete EpiLife medium every other day. When cell outgrowth reached 80% confluence, the p0 keratinocytes were detached with 0.025% trypsin and ethylenediaminetetraacetic acid (Thermo Fisher Scientific, Waltham, MA, USA), neutralized with defined trypsin inhibitor (Thermo Fisher Scientific, Waltham, MA, USA), and replated as p1 cells into another tissue-culture Petri dish ($1.0 \times 10^4$ cells/cm$^2$) with complete EpiLife medium. Passage 1 (p1) keratinocytes obtained from the skin, vermilion, and oral mucosa were fed with the same culture medium every other day. After a confluence of approximately 80% was reached, the total RNA was extracted from the cells using a RNeasy mini kit (Qiagen, Valencia, CA, USA), according to the manufacturer's instructions.

## 2.4. Microarray Analysis

The microarray data were deposited into the NCBI Gene Expression Omnibus Database (GSE 172126). The labeled aRNA was fragmented and hybridized to an Affymetrix GeneChip Cynomolgus + Rhesus Gene 1.0 ST array (Cat# 901941), which covers 5319 and 37,293 genes of Cynomolgus and Rhesus monkeys, respectively. The signals detected for each gene were normalized using the Robust Multi-array Average algorithm.

Microarray data were analyzed using a microarray data analysis tool (Filgen, Nagoya, Japan). To exclude data with low reliability, following normalization, the probe sets were excluded, which showed expression values below the average expression value of 2318 negative control probes. These were putative intronic-based probe sets from 100 adult putative housekeeping genes. Then, six pairwise comparisons were performed with keratinocytes in the vermilion, oral mucosa, and epidermis in vivo and in vitro. The expression values for vermilion and epidermal keratinocytes were always used as test and control samples, respectively. After the ratio of expression value had been calculated between the test and control sample, it was converted to log2 ratio, which was referred to as logFC. Then, the up-regulated and down-regulated genes were determined by the test sample's expression values with a logFC of | ≥1 | compared with the control, respectively. This resulted in 2059 genes, which were filtered by applying them to at least one or more up-regulated or down-regulated genes. To generate a heat map, the logFC values of

filtered genes were clustered using R software's Euclidean distance and Ward's method (http://www.r-project.org/ (accessed on 29 November 2021)).

*2.5. Histological and Immunohistochemical Examination of Lip Tissue from a Japanese Monkey and Human*

The lip tissue of a Japanese macaque not used for cell culture was fixed with 4% paraformaldehyde in 100 mM D-PBS and embedded in paraffin for the histological examination. The paraffin-embedded samples were deparaffinized, rehydrated, cut into 5-μM thick sections, and stained with hematoxylin and eosin (H & E) for histological examination. In addition, the sections were prepared for immunohistochemical examination of keratin 10 (K10) and small proline-rich protein 3 (SPRR3). The sections were deparaffinized in xylene and rehydrated in ethanol. Endogenous peroxidase was blocked with 0.3% hydrogen peroxide in methanol for 30 m. Antigen retrieval was achieved by autoclaving with 10 mM citric sodium buffer (pH 6.0) at 100 °C for 20 m. After incubating with 5% BSA in phosphate-buffered saline (PBS) for 30 m, sections were incubated with either a mouse monoclonal antibody against K10 (ab9026) (Abcam, Cambridge, UK) at 1:200 or a mouse monoclonal antibody against SPRR3 (ab58233) (Abcam, Cambridge, UK) at of 1:100, at 4 °C overnight. After washing with TBS, the sections were reacted with EnVision FLEX Plus (Dako, Carpinteria, CA, USA) at room temperature for 1 h. The immunoreactions were then visualized with 3,3'diaminobenzidine (Dojindo Co. Ltd., Kumamoto, Japan), and the sections were counterstained with hematoxylin.

For the histological examination of the human lip (female, 53-year-old), frozen whole lip tissue was purchased and imported from Science Care (Phoenix, AZ, USA). The donor died of acute myeloblastic leukemia. The median part of the lower lip was excised and embedded in SCEM (SECTION-LAB, Hiroshima, Japan), cut into 6 μm sections, and fixed with methanol at −30 °C for 10 m. The sections were washed with D-PBS three times and stained with H & E. For immunohistochemical examination, the sections were incubated in a blocking buffer (1% new-born calf serum in D-PBS) for 60 m at room temperature and incubated with primary antibodies against K10, SPRR3, and blocking buffer overnight at 4 °C. Then, the samples were washed three times with D-PBS and incubated with secondary antibodies for 60 m at room temperature. After washing, the nuclei were stained with DAPI (1:700, D1306, Thermo Fisher Scientific, Waltham, MA, USA) for 1 h and mounted. The primary antibodies that were used and their dilution rates were: guinea pig anti-K10 (GP-K10) (1:200, Progen, Heidelberg, Germany) and rabbit anti-SPRR3 (HPA044467) (1:1000, Atlas Antibodies, Bromma, Sweden). Secondary antibodies (Alexa 488 (A-11070) and 647 (A-21450, Thermo Fisher Scientific, Waltham, MA, USA) were used at 1:1000 dilution.

### 3. Results

*3.1. Histological Characteristics of Japanese Macaques' Lips and Establishment of Primary Keratinocyte Culture Derived from SkinVvermilion/Oral Mucosa within the Lip*

Microscopic observations in this study revealed that 'vermilion epithelium' was histologically present in Japanese macaques' lips, which is similar to that in humans (Figure 1a,b). In the macaques, it was mainly a parakeratinized stratified epithelium with prominent rete ridges devoid of any appendages in the underlying tissue. This encouraged us to further examine the characteristics of vermilion epithelium using Japanese macaques' lips. Since Sudan black allowed us to macroscopically distinguish vermilion from oral mucosa within the lip epithelium (Figure 2), the staining facilitated the successful separation of the skin, vermilion, and oral mucosa of the lip. Also, we were able to establish monkey primary keratinocytes culture and serial cell culture using a serum-free, complete EpiLife (Thermo Fisher Scientific, Waltham, MA, USA) medium, identical to our primary human oral keratinocyte culture system. However, there were little morphological differences among the cultured keratinocytes harvested from the skin, vermilion, and oral mucosa (Supplementary Materials Figure S1a–c).

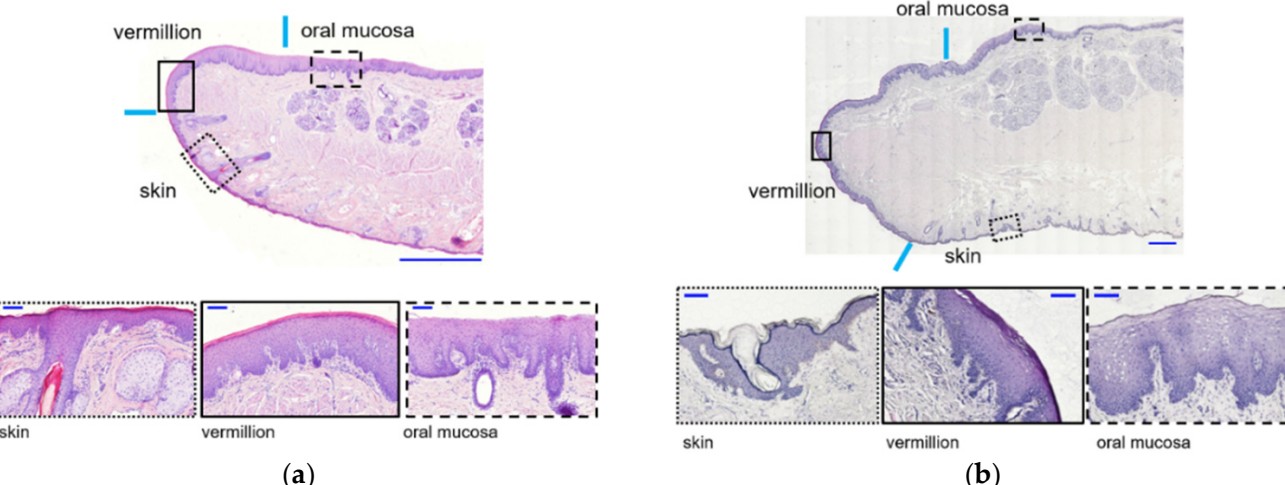

(**a**)                                                                                          (**b**)

**Figure 1.** Histological and immunohistochemical appearances of lip tissue of a Japanese monkey and a human: (**a**) histological appearance of monkey lip. (H & E staining) Light blue bars indicate a possible histological border between vermilion epithelium and the adjacent epidermis/oral mucosa, respectively. Original magnification ×10. Scale bar = 2000 μm. Three panels on the bottom are enlargements of the corresponding insets delineated with dotted (skin), solid (vermilion) and dashed (labial oral mucosa) lines on the top panel. Original magnification ×40. Scale bar = 200 μm; (**b**) histological appearance of human lip. (H & E staining) Light blue bars indicate a possible histological border between vermilion epithelium and the adjacent epidermis/oral mucosa, respectively. Original magnification ×10. Scale bar = 1000 μm. Three panels on the bottom are enlargements of the corresponding insets delineated with dotted (skin), solid (vermilion), and dashed (labial oral mucosa) lines on the top panel. Original magnification ×40. Scale bar = 200 μm.

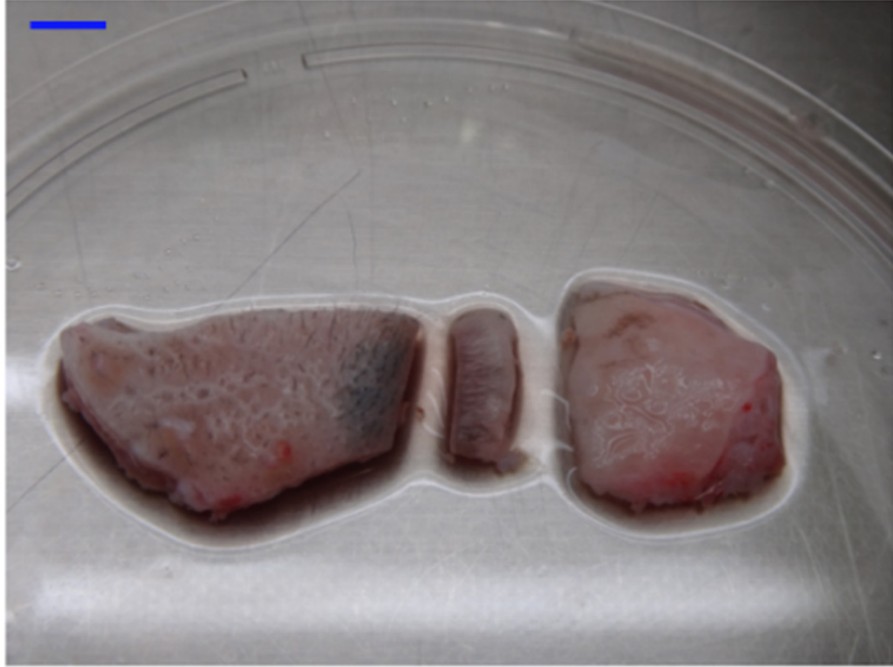

**Figure 2.** Macroscopic image of three distinct tissue specimens (skin, vermilion, and oral mucosa) of the monkey lip after separating especially by determining between vermilion and adjacent oral mucosa. Scale bar = 1 cm.

### 3.2. Gene Expression Profile of the Skin, Vermilion and Oral Mucosa of the Lip In Vivo and In Vitro

Next, we conducted microarray analysis to examine the gene expression profile in Japanese macaques' lips for finding differentially expressed genes specific to vermilion keratinocytes by comparing it with keratinocytes from the lip epidermis and oral mucosa in vivo and in vitro. As a result, we found the 2059 genes were differentially up-regulated or down-regulated in pairwise comparison among six groups, and then we generated a heat map using them (Figure 3, Tables 1 and S1). The results indicated that the gene expression profile of vermilion keratinocytes in vivo resembled those in vitro. Also, the gene expression profile of epidermal keratinocytes in vivo appeared to be very different from the vermilion and oral mucosa keratinocytes in vivo. In addition, the epidermal keratinocytes in vitro were similar to the vermilion and oral mucosa keratinocytes in vitro. Furthermore, we categorized the genes into 15 clusters (Figure 3). Their expression levels within clusters C11–15 and C3, 4, 6, 11, and 12 were relatively higher in vermilion keratinocytes in vivo than in epidermal and oral keratinocytes, respectively, providing several differentially expressed genes.

### 3.3. A Potential Marker Specific to Vermilion Epithelium by Immunohistochemical Staining

Since the 69 genes clustered into C11 were up-regulated only in vermilion in vivo, they could be a single potential marker specific to vermilion epithelium. There are 14 genes more up-regulated among them (Table 1). Therefore, to support the gene expression profile specific to vermilion keratinocytes, the expression level of repetin (RPTN), serine peptidase inhibitor kazal type 9 (SPINK9), keratin 222, and keratin 2B (=keratin 76) was immunohistochemically analyzed in both human and monkey lip in vivo. However, their antibodies to humans did not specifically react with the vermilion epithelium (figures not shown). We then chose one highly-expressed gene from each of C6 and C13 cluster, which is K10 and SPRR3, a cornified cell envelope precursor, and examined a double-immunostaining. In the monkey lip epithelium, K10 expression was observed in the epidermal suprabasal layer and diminished across the vermilion. Then, it was lost as the vermilion epithelium changed from orthokeratinized to parakeratinized, and absent in the oral mucosa (Figure 4a). In contrast, the SPRR3 was expressed in the suprabasal layer of the oral mucosa. Its expression within the vermilion epithelium extended toward the skin but was gradually restricted to the superficial layer and disappeared in the middle of vermilion when orthokeratinized. Finally, it was absent in the epidermis (Figure 4b). This inverse immunostaining pattern resembles human lips, and K10 and SPRR3 expression overlapped within the vermilion (Figure 4c–e). As a result, this study found that the combination of K10 and SPRR3 can be used as a potential marker of vermilion epithelium in human lip.

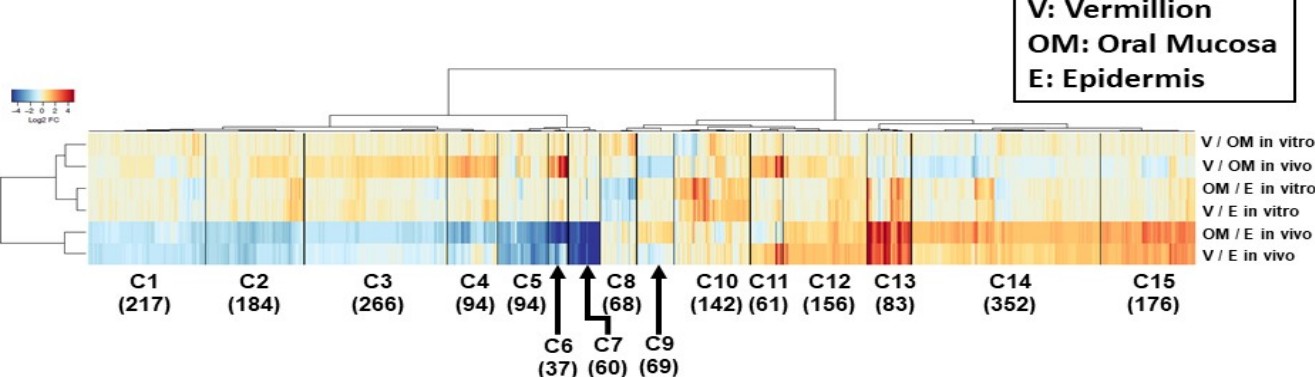

**Figure 3.** Clustering analysis of 2059 genes expressed in keratinocytes of vermilion, oral mucosa, or epidermis in vivo and in vitro. Clusters (C1–15) of genes expressed by pairwise comparison with keratinocytes of vermilion, oral mucosa, or epidermis in vivo and in vitro. Orange and blue denote up- and down-regulation, respectively. The color bar shows a gradient of fold-changes. Numbers of clustered genes are denoted in round brackets.

**Table 1.** C11 and C13 with a logFC of ≥2. logFC shown are vermilion versus oral mucosa in vivo in C6, vermilion versus epidermis or oral mucosa in vivo in C11, and vermilion versus epidermis in vivo in C13. Names of the genes shown in bold were immune-stained as associated proteins in Figure 3. Full gene description of the gene symbol in this table is listed in Supplementary Materials Table S2. The microarray data were deposited into the NCBI Gene Expression Omnibus Database (GSE 172126).

| C6 | | C11 | | | | C13 | | | |
|---|---|---|---|---|---|---|---|---|---|
| Gene Symbol | Log FC (vs. OM Vivo) | Gene Symbol | Log FC (vs. OM In Vivo/E In Vivo) | Gene Symbol | Log FC (vs. E In Vivo) | Gene Symbol | Log FC (vs. E In Vivo) | Gene Symbol | Log FC (vs. E In Vivo) |
| GSDMA | 2.2953 | ABCC5 | 2.0328/1.1615 | GBP6 | 3.5856 | MIR147B | 3.7021 | SPINK7 | 2.8136 |
| TMEM45A | 2.7063 | RPTN | 7.8735/4.4448 | KRT4 | 8.4516 | GALNT12 | 2.9896 | CRNN | 4.3622 |
| LOC100426523 | 2.2847 | KRT2B | 7.3961/4.7954 | **LOC713857 (SPRR3)** | 8.7096 | LOC713440 | 3.0797 | SERPINB11 | 4.0852 |
| POSTN | 2.2028 | LOC701667 | 7.0472/4.9698 | TMPRSS11D | 5.2985 | MBOAT1 | 2.1566 | LOC717546 | 3.3945 |
| LOC695550 | 2.3715 | LOC722692 | 2.3923/2.5363 | IL1F6 | 6.1944 | NT5E | 2.5473 | PAX9 | 3.4938 |
| WFDC5 | 2.2929 | LOC717970//LOC717993 | 2.9361/3.1381 | CRISP3 | 8.7735 | GPRC5A | 2.7553 | CEACAM5 | 3.3865 |
| LOC100423330 | 2.1396 | LOC710890 | 2.9022/1.5895 | CXCL17 | 5.8785 | SCNN1G | 3.1839 | SOX2 | 2.9894 |
| DSC1 | 3.9815 | PNLIPRP3 | 2.5550/1.5605 | LOC714502 | 6.3552 | LOC100426646 | 3.6829 | GCNT1 | 2.1485 |
| LOC712658 | 4.6311 | DNER | 2.2476/1.8572 | TMPRSS11B | 6.6002 | SAMD9 | 4.4959 | PITX1 | 2.3245 |
| BPIL2 | 4.1798 | LOC100425096 | 3.6931/4.6541 | ADA | 6.5452 | S100A9 | 5.2672 | A2ML1 | 2.4604 |
| LOC722683 | 4.8427 | LOC100423543 | 3.9080/2.3828 | ECM1 | 2.6198 | CRYGS | 4.8682 | EHD3 | 2.1601 |
| SERPINA12 | 3.0485 | SPRR2G | 4.0516/2.4677 | LOC719353 | 3.6185 | ETNK2 | 5.6011 | KRT78 | 2.5619 |
| LOC713408 | 3.0586 | KRT222 | 4.1171/3.2877 | SLC16A9 | 3.1161 | CST2 | 3.9815 | LOC713909 | 2.5648 |
| KLK5 | 3.3683 | SPINK9 | 5.3136/3.9029 | ATP13A5 | 2.6948 | CST1 | 3.9212 | FTH1//LOC699053 | 2.1233 |
| CASP14 | 3.2255 | | | PCBD1 | 2.5473 | LOC722514 | 3.7479 | ST6GALNAC1 | 2.2786 |
| PM20D1 | 3.1205 | | | SLPI | 3.3339 | TRPM2 | 3.7346 | ABCA6 | 2.4280 |
| SERPINB12 | 3.2568 | | | POPDC3 | 2.7387 | LOC704291 | 3.8116 | SLC1A4 | 2.7600 |
| ASPRV1 | 3.8937 | | | GPR160 | 2.9576 | ADH1C | 4.2836 | LOC713400 | 2.8427 |
| LOC702153 | 4.9829 | | | LOC706293 | 3.1363 | AKR1B10 | 4.1351 | SLC25A21 | 3.2338 |
| CDSN | 4.5466 | | | LOC696056 | 3.0656 | SERPINB4 | 3.4467 | CLDN17 | 3.3863 |
| LOC100425304 | 6.6358 | | | VIL1 | 3.0458 | SLURP1 | 2.5486 | LOC722294 | 3.1881 |
| FLG2 | 6.2283 | | | RANBP3L | 2.8945 | SLC6A15 | 2.7791 | S100A8 | 3.1013 |
| KRT2 | 6.4523 | | | LOC715292 | 3.0353 | PITX2 | 2.4275 | LOC717871 | 3.3628 |
| **KRT1 (K10)** | 7.0481 | | | CLDN7 | 3.0907 | PAPSS2 | 3.4387 | | |
| | | | | IL1A | 3.0709 | LOC695158 | 2.9003 | | |

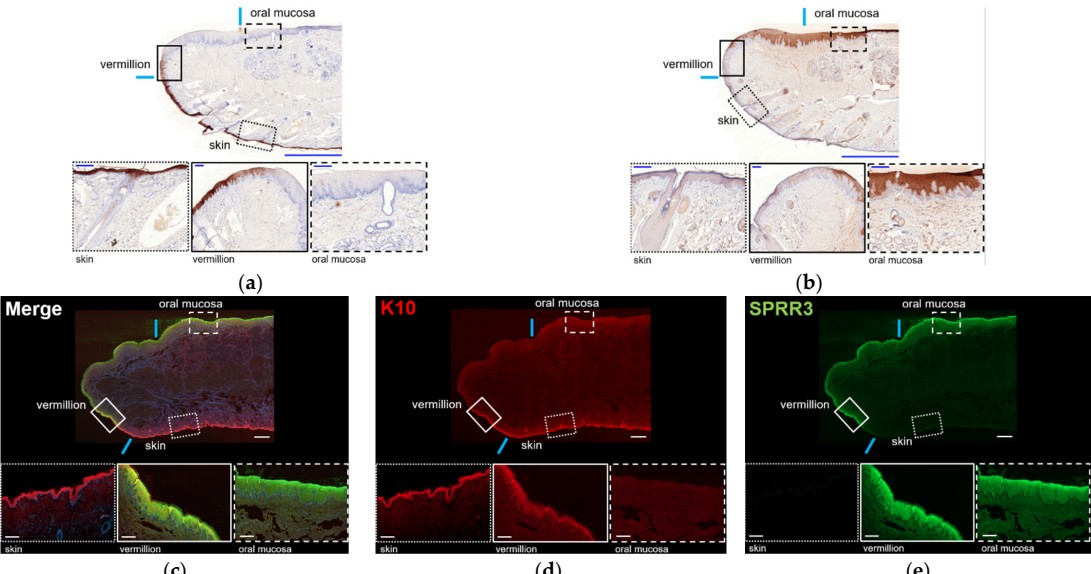

**Figure 4.** (**a**) K10 immunohistochemistry of monkey lip. Light blue bars indicate a possible histological border between vermilion epithelium and the adjacent epidermis/oral mucosa, respectively. Original

magnification ×10. Scale bar = 2000 µm. Three panels on the bottom are enlargements of the corresponding insets delineated with dotted (skin), solid (vermilion), and dashed (labial oral mucosa) lines on the top panel. Original magnification × 20. Scale bar = 200 µm; (**b**) SPRR3 immunohistochemistry of monkey lip. Light blue bars indicate a possible histological border between vermilion epithelium and the adjacent epidermis/oral mucosa, respectively. Original magnification ×10. Scale bar = 2000 µm. Three panels on the bottom are enlargements of the corresponding insets delineated with dotted (skin), solid (vermilion), and dashed (labial oral mucosa) lines on the top panel. Original magnification ×20. Scale bar = 200 µm; (**c**) double immunofluorescence staining of K10 (red) and SPRR3 (green) in the human lip. The specimen was counterstained with DAPI (blue). Light blue bars indicate a possible histological border between vermilion epithelium and the adjacent epidermis/oral mucosa, respectively. Original magnifications ×10. Scale bar = 1000 µm. Three panels on the bottom are enlargements of the corresponding insets delineated with dotted (skin), solid (vermilion), and dashed (labial oral mucosa) lines on the top panel. Original magnification × 20. Scale bar = 200 µm; (**d**) immunofluorescence staining of K10 (red) in the human lip. Light blue bars indicate a possible histological border between vermilion epithelium and the adjacent epidermis/oral mucosa, respectively. Original magnifications ×10. Scale bar = 1000 µm. Three panels on the bottom are enlargements of the corresponding insets delineated with dotted (skin), solid (vermilion) and dashed (labial oral mucosa) lines on the top panel. Original magnification ×20. Scale bar = 200 µm; (**e**) immunofluorescence staining of SPRR3 (green) in the human lip. Light blue bars indicate a possible histological border between vermilion epithelium and the adjacent epidermis/oral mucosa, respectively. Original magnifications ×10. Scale bar = 1000 µm. Three panels on the bottom are enlargements of the corresponding insets delineated with dotted (skin), solid (vermilion), and dashed (labial oral mucosa) lines on the top panel. Original magnification × 20. Scale bar = 200 µm.

## 4. Discussion

The vermilion epithelium's histological characteristics are distinct from the adjacent epidermis and oral mucosa in human lips; however, our findings demonstrated that those in the monkey lip were similar to humans, despite the fact that vermilion is known to be inherent in *Homo sapiens* [1,10]. These findings could help us understand the vermilion epithelium's unique differentiation and keratinization. Furthermore, it could lead to the discovery of specific epithelial markers for vermilion and might facilitate the development of a human lip/vermilion model as a tool for quality assurance of therapeutic products of lips [8].

Nonetheless, it is unclear why the Japanese macaque has vermilion similar to histological features of the human lip. A recent hypothetical study revealed that non-aggressive friction occurring in the lip may decrease the density of hair follicles in human skin, suggesting the relationship between pronunciation and the histological appearance of the vermillion [11]. Although Chimpanzees do not speak, the lip movement associated with facial expressions and social functions, which is not as much as that in human, can produce mild mechanical stimuli to the vermillion. This could be due to the similarities of vermilion of human and monkey [12]. According to a previous report, the porcine snout has an area with absence of adnexal structures like the human lips, which is a distinct characteristic relative to other porcine and human skin [4]. Altogether, long-term mild friction to the perioral area may play a role, in part, in vermillion histology development.

The key issue of this study was to macroscopically determine the border between vermilion and skin/oral mucosa, which can avoid major contamination of keratinocytes derived from other different epithelial tissues. To achieve it, we devised a method to especially distinguish vermilion from the oral mucosa within monkey lip. Vermilion surface lipids originate from blood vessel penetration, not sebum secreted from sebaceous glands, which demonstrates the lipophilicity of vermilion [13]. Sudan black B is used to stain various lipids such as phospholipids, sterols, and neutral triglycerides. Therefore, because Sudan black dye rendered the border between hairless vermilion and oral mucosa discriminable, it was successful to separate the skin, vermilion, and oral mucosa within the lip epithelium. Additionally, this technique facilitated the development of monkey

primary keratinocytes in culture, possibly for the first time, despite the use of the explant culture technique, which often causes fibroblast contamination. This could be due to the primary keratinocyte culture system of serum-free and low $Ca^{++}$ (0.06 mM) conditions, identical to the human clinical application [14]. Conversely, because this culture condition maintains cultured keratinocytes in a proliferative and undifferentiated state [15], there were little morphological differences among the cultured keratinocytes harvested from the skin, vermilion, and oral mucosa. Further studies are required to develop an in vitro lip model that exhibits the in vivo lip keratinocytes' profile.

Our clustering analysis demonstrated differences in the vermilion between the skin and oral mucosa within the lip of a Japanese macaque, although the statistical power was limited. First, the gene expression profile of vermilion keratinocyte in vivo was relatively similar to that of any keratinocytes in vitro. This suggests a more proliferative potential of vermilion epithelium, which is important to develop an in vitro model. Second, in vivo, the profile of skin epithelium was much different from that of the vermilion. This could be attributed to the presence of hair and terminal differentiation in the epidermis. Third, more importantly, the expression levels of genes clustered into C11 was highly up-regulated in vermilion keratinocytes in vivo. Therefore, RPTN, SPINK9, keratin222, and keratin2B were expected to be single potential markers specific to vermilion. However, our results failed to indicate that the microarray data were consistent with the immunohistochemical findings, although the increase in mRNA levels did not necessarily reflect the same in the protein level. Nonetheless, all RPTN, SPINK9, keratin222, and keratin2B have an important and unique role in desquamation process and epithelial differentiation in squamous epithelium. Therefore, these proteins associated with genes in the C11 cluster require further studies to explore the specific functions of vermilion as well as a potential marker.

Instead, we focused on K10 and SPRR3 categorized into C6 and C13, respectively, because the proteins associated with these clustered genes could be used as combination markers to distinguish vermilion epithelium from the epidermis and oral mucosa. Additionally, previous studies on developing lip in vitro models reported their immunoreaction independently [16,17]. Considering that gene expression levels of K10 and SPRR3 agreed with the associated protein expression and localization in human lips, we concluded that K10 and SPRR3 can be used as specific double combination markers for vermilion epithelium in human, although their expression pattern was a little different from that in the monkey lip.

Vermilion's phenotypic features between the adjacent epidermis and oral mucosa in the lip are not well-known, despite its distinct characteristics. The expression levels of terminal differentiation markers in stratified squamous epithelia, such as loricrin, involucrin, and SPRRs, are regulated and triggered by various intrinsic factors and internal and external stimuli, such as the underlying connective tissue and moist conditions [18–20]. Also, the functional assay to measure transepithelial water loss suggested incomplete cornified layer formation of the lip surface [6]. Moreover, attention should be paid to lipid metabolism and cholesterol sulphate in vermilion and crosstalk through soluble factors from the underlying tissue [21,22]. Further investigation is required to determine the three distinct epithelia within the lip to facilitate the development of a human in vitro model to understand unique lip biology.

We recognize that the very small sample size is a limitation of this study, although harvesting the entire lip tissue from humans and monkeys is very challenging due to cosmetic and ethical issues. Additionally, because our immunohistochemical analysis to verify the protein expression specific to vermillion epithelium did not provide any significant results, this incompletion to demonstrate a single marker was another weak point of this study. It may be required to produce higher quality antibodies for immunostaining or to perform in situ hybridization clustered in C11 to overcome this state.

## 5. Conclusions

Although the microarray analysis of this study could not lead to the detection of a single marker specific to vermilion epithelium based on the gene expression profile of a Japanese macaque, the pair of K10 and SPRR3 resulted in a potential marker of vermilion epithelium in the human lip.

**Supplementary Materials:** The following supporting information can be downloaded at: https://www.mdpi.com/article/10.3390/anatomia1010002/s1, Figure S1a–c: Representative microscopic images of p1 keratinocytes derived from three distinct epithelia (a: skin, b: vermilion, c: oral mucosa) within the lip of Japanese macaques; Table S1: List of 2059 differentially up-regulated or down-regulated genes in six pairwise comparisons. Table S2: List of full gene descriptions shown in Table 1.

**Author Contributions:** Conceptualization, H.K. and K.I.; methodology, H.K., E.H., A.S., K.H., E.N., and K.I.; software, Y.L. and S.O.; validation, H.K., S.O. and K.I.; formal analysis, H.K., S.O. and K.I.; investigation, H.K., E.H., A.S., K.H., E.N., and A.U.; resources, K.I.; data curation, Y.L. and S.O.; writing—original draft preparation, H.K. and K.I.; writing—review and editing, Y.L., E.H., A.U., E.N., S.O. and K.I.; visualization, H.K., Y.L. and S.O.; supervision, K.I.; project administration, K.I.; funding acquisition, A.U. All authors have read and agreed to the published version of the manuscript.

**Funding:** This research was funded by JSPS KAKENHI Grant Number 19K19068 to A.U.

**Institutional Review Board Statement:** We harvested the entire upper and lower lip tissues from one male Japanese macaque monkey (*Macaca fuscata*) (8.2 kg) (7 years and 9 months old), provided by the NBRP-Nihonzaru at Kyoto University Primate Research Institute with support in part by the National Bio-Resource Project of the Japan Agency for Medical Research and Development (AMED). This experiment complied with the National Institutes of Health Guidelines for the Care and Use of Laboratory Animals. The animal study protocol was approved by the Niigata University Institutional Animal Care and Use Committee (protocol code of SA00008, 31 March 2017).

**Informed Consent Statement:** Not applicable.

**Data Availability Statement:** The microarray data that support the findings of this study are openly available in [the NCBI Gene Expression Omnibus Database] at [https://www.ncbi.nlm.nih.gov/ (accessed on 4 January 2022)], reference number [GSE172126].

**Acknowledgments:** The authors are grateful to Isao Hasegawa and associate Keisuke Kawasaki and staff member of the Department of Physiology, Niigata University School of Medicine, for their technical assistance during harvesting tissues from Japanese macaques.

**Conflicts of Interest:** The authors declare no conflict of interest.

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
