# Peer review of "Detection of Potential Markers for Lip Vermilion Epithelium in Japanese Macaques Based on the Results of Gene Expression Profile"

_2813-0545, doi:10.3390/anatomia1010002_

Round 1

Reviewer 1 Report

Dear Authors,

In this study, the authors aimed “to perform microarray analysis for in vivo tissues as well as primary keratinocytes in a culture derived from three distinct epithelia in the lip of a Japanese macaque, resembling histological characteristics of the human lip, and examine the gene expression profile that distinguished vermilion epithelium from adjacent skin and oral mucosa epithelia”. According to my assessments, I've thought that the current study could be pertinent and informative for the readers of “Anatomia”. In my opinion, it is organized and written well. Furthermore, the findings of current vigorous study could be helpful for ongoing studies investigating specific epithelial markers for vermilion.

Besides, according to my assessments, there are some minor points to be revised and clarified. Please revise and clarify following issues;

  1. In my opinion, the abstract seem concise. It sufficiently describes the study. In addition, I think that the ages, and genders of the samples (Japanese macaque, human) included into the study could be given in the abstract, and also M&M sections.
  2. In the introduction section, 8th sentence of the 1st paragraph should have been cited.
  3. In the M&M section, rationale of usage, administration way and dose for “Sudan Black B staining” used in to differentiate the vermillion area should be given clearly. In addition, as well as the manufacturers' names, their city and country of origin for all materials, including this agent, used in the study should be specified in brackets.
  4. In the M&M section, in the 5th sentence of 5th paragraph, the manufacturer name, and its city and country of origin for "defined trypsin inhibitor" used in the study should be specified in brackets.
  5. In the "3.3. A Potential Marker Specific........" part of the M&M section, in the 2nd sentence, the number of the "Table" should have been specified in the bracket.
  6. In the Table1, Cluster of C13 given in the table legend does not seem consistent with the cluster C12 given in table 1. Also, in my opinion, "names of the genes shown in bold" in the Table 1 should be revised, seems missing.
  7. In the discussion section, in my opinion, it could be discussed why Japanese macaque had vermilion resembling histological characteristics of the human lip, despite the fact that vermilion is known to be inherent in Homo sapiens.

As a conclusion, according to my assessments in general, I think that there is sufficient and valuable addition to the current literature in the current study, in order to consider publishing in Anatomia.

Author Response

Responses to Reviewer #1

  1. In my opinion, the abstract seem concise. It sufficiently describes the study. In addition, I think that the ages, and genders of the samples (Japanese macaque, human) included into the study could be given in the abstract, and also M&M sections.

Thank you for your comment. Those information has been added.

  1. In the introduction section, 8th sentence of the 1st paragraph should have been cited.

Thank you for your comment. One paper was cited.

  1. In the M&M section, rationale of usage, administration way and dose for “Sudan Black B staining” used in to differentiate the vermillion area should be given clearly. In addition, as well as the manufacturers' names, their city and country of origin for all materials, including this agent, used in the study should be specified in brackets.

Thank you for your comments. We agreed with you. As suggested, we included those information in the sections of M&M and Discussion, separately.

  1. In the M&M section, in the 5th sentence of 5th paragraph, the manufacturer name, and its city and country of origin for "defined trypsin inhibitor" used in the study should be specified in brackets.

We apologized for the missing information, and it was included in the revised manuscript.

  1. In the "3.3. A Potential Marker Specific........" part of the M&M section, in the 2nd sentence, the number of the "Table" should have been specified in the bracket.

We apologized for it, and added the number.

  1. In the Table1, Cluster of C13 given in the table legend does not seem consistent with the cluster C12 given in table 1. Also, in my opinion, "names of the genes shown in bold" in the Table 1 should be revised, seems missing.

We apologized for the errors. We corrected both.

  1. In the discussion section, in my opinion, it could be discussed why Japanese macaque had vermilion resembling histological characteristics of the human lip, despite the fact that vermilion is known to be inherent in Homo sapiens.

Thank you for your valuable comments and suggestions. As suggested, we added some statements with citing refereances in the revised manuscript.

Reviewer 2 Report

Dear Authors,

we really need such a manuscripts, thank you.

However, I have some objections and would like to invite the authors to address them for the improvement of the manuscript:

1) Introduction. Too short and doesnt give common characterization of the types in epithelium, including vermilion, of the lip! Please, specify the morphological characterization of lip epithelia, focus on the known positive tissue factors and most characteristic for each type. This will not only improve the Introduction and give the info to the reader, but also seriously expand the References part, what is very scarce now.

I have also one remark here (for the last paragraph) - IMH doesnt measure the level of factors, but give just the idea about the factors appearance, please, be precise in yours description!

2) Materials and methods. Please, bring also the year of issue of Ethical Committee permission in the main text;

in 2.5 give please full title of the antibodies used and only after the abbreviations used;

Line 150-152, please add the age, sex, inclusion/exclusion criteria (skin disease, medicine used, time of the death - summer, winter etc) for the human subject used for your work. This is an important info as lip react on many environmental, also internal factors...

3) Results. Commonly nice Figs except the upper review picture of lip for Fig 1b! If your scanner cant take the picture properly (and such situation might appear, I know!), please, use the quality improving Programs. I would advice to go for paint.net, download free of charge and correct, please, the small boxes there, but you cant leave this picture how it is. Perhaps it is possible to use an other scanner, get the tif expansion first with following transfer in jpg...!?

Table 1. Here you have to decipher all titles of the genes used and add as an Attachment File or at the end of the manuscript.

4) Discussion. Please, include the paragraph of Limitations at the end.

5) Separate Conclusions, please.

6) References are highly not enough. Please, expand the Introduction and Youll be able to expand this part, too!

Author Response

Replies to Reviewer #2

1) Introduction. Too short and doesnt give common characterization of the types in epithelium, including vermilion, of the lip! Please, specify the morphological characterization of lip epithelia, focus on the known positive tissue factors and most characteristic for each type. This will not only improve the Introduction and give the info to the reader, but also seriously expand the References part, what is very scarce now.

I have also one remark here (for the last paragraph) - IMH doesnt measure the level of factors, but give just the idea about the factors appearance, please, be precise in yours description!

Thank you veru much for your comments that should be a great input to improve our manuscript. We described more detailed information in the section of Introduction with an increase in the number of references. Additionally, we rephrased “protein expression levels”.

2) Materials and methods.

Please, bring also the year of issue of Ethical Committee permission in the main text;

in 2.5 give please full title of the antibodies used and only after the abbreviations used;

Thank you for your comment. Those information have been added.

Line 150-152, please add the age, sex, inclusion/exclusion criteria (skin disease, medicine used, time of the death - summer, winter etc) for the human subject used for your work. This is an important info as lip react on many environmental, also internal factors...

Thank you for your comment. We understand your point. However, this is not a human clinical study. In fact, the human lip tissue used in this study was a type of merchandise. When it was ready to ship after we made an order, the company in the US sent it to us. Therefore, there are no inclusion/exclusion criteria for the lip tissue used in this study.

3) Results. Commonly nice Figs except the upper review picture of lip for Fig 1b! If your scanner cant take the picture properly (and such situation might appear, I know!), please, use the quality improving Programs. I would advice to go for paint.net, download free of charge and correct, please, the small boxes there, but you cant leave this picture how it is. Perhaps it is possible to use an other scanner, get the tif expansion first with following transfer in jpg...!?

We appreciate your comment and providing the detailed information. As abovementioned to the Academic Editor, we replaced it by new images.

Table 1. Here you have to decipher all titles of the genes used and add as an Attachment File or at the end of the manuscript.

We thank your comment and suggestions, and we agreed with you. As another supplementary material as a new Table S2, we added the list of full gene descrptions shown in the original Table 1.

4) Discussion. Please, include the paragraph of Limitations at the end.

Thank you very much for your comment. As suggested, we have included it in the revised manuscript.

5) Separate Conclusions, please.

Thank you for your comment. As suggested, we did it in the revised manuscript.

6) References are highly not enough. Please, expand the Introduction and Youll be able to expand this part, too!

Thank you for your comment. As stated above, we included references as many as possible.